# Evolution of SARS-CoV-2 Spikes shapes their binding affinities to animal ACE2 orthologs

Weitong Yao,[1,2,3] Yujun Li,[2] Danting Ma,[2,4] Xudong Hou,[1,2] Haimin Wang,[5] Xiaojuan Tang,[1,2] Dechun Cheng,[2,6] He Zhang,[1,2] Chengzhi Du,[1,2] Hong Pan,[1,2] Chao Li,[2] Hua Lin,[7] Mengsi Sun,[2] Qiang Ding,[8] Yingjie Wang,[2] Jiali Gao,[1,2,9] Guocai Zhong[1,2,5]

**ABSTRACT**  Spike-receptor interaction is a critical determinant for the host range of coronaviruses. Here, we investigated all the five World Health Organization-designated variants of concern (VOC), including Alpha (B.1.1.7), Beta (B.1.351), Gamma (P.1), Delta (B.1.617.2), and Omicron (B.1.1.529), for their Spike receptor-binding domain (RBD)'s interactions with ACE2 orthologs of 18 animal species. We found that, compared to the RBD of an early isolate WHU01, the Alpha RBD has markedly increased affinity to cattle and pig ACE2 proteins and decreased affinity to horse and donkey ACE2 proteins. The RBDs of Beta and Gamma variants have almost completely lost affinity to bat, horse, and donkey ACE2 orthologs. Mainly due to the Q493R and N501Y mutations, the Omicron RBD showed markedly enhanced affinity to mouse ACE2. Molecular dynamic simulations further suggest that Omicron RBDs are optimal for electrostatic interactions with mouse ACE2. Interestingly, the Omicron RBD also showed decreased or complete loss of affinity to eight tested animal ACE2 orthologs, including that of horse, donkey, pig, dog, cat, pangolin, American pika, and bat. The K417N, G496S, and Y505H substitutions were identified as three major contributors that commonly have negative impact on RBD binding to these eight ACE2 orthologs. These findings show that Spike mutations have been continuously shaping SARS-CoV-2's binding affinities to animal ACE2 orthologs and suggest the importance of surveillance of animal infection by circulating SARS-CoV-2 variants.

**IMPORTANCE**  Spike-receptor interaction is a critical determinant for the host range of coronaviruses. In this study, we investigated the SARS-CoV-2 WHU01 strain and five WHO-designated SARS-CoV-2 variants of concern (VOCs), including Alpha, Beta, Gamma, Delta, and the early Omicron variant, for their Spike interactions with ACE2 proteins of 18 animal species. First, the receptor-binding domains (RBDs) of Alpha, Beta, Gamma, and Omicron were found to display progressive gain of affinity to mouse ACE2. More interestingly, these RBDs were also found with progressive loss of affinities to multiple ACE2 orthologs. The Omicron RBD showed decreased or complete loss of affinity to eight tested animal ACE2 orthologs, including that of some livestock animals (horse, donkey, and pig), pet animals (dog and cat), and wild animals (pangolin, American pika, and *Rhinolophus sinicus* bat). These findings shed light on potential host range shift of SARS-CoV-2 VOCs, especially that of the Omicron variant.

**KEYWORDS**  SARS-CoV-2, variant of concern, Omicron, ACE2, host range, mouse, molecular dynamics simulations

The coronavirus disease 2019 (COVID-19) pandemic has triggered unprecedentedly extensive worldwide efforts to develop countermeasures against COVID-19, and a number of progresses have been achieved in developing anti-SARS-CoV-2 prophylactic vaccines, antibody biologics, and small molecule drugs (1–16). So far, a number

Address correspondence to Yujun Li, liyujun@szu.edu.cn, Weitong Yao, yaowt@hbjxlab.com, Yingjie Wang, wangyj@szbl.ac.cn, Jiali Gao, gao@jialigao.org, or Guocai Zhong, Guocai.Zhong@umassmed.edu.

Weitong Yao, Yujun Li, and Danting Ma contributed equally to this article. Author order was determined by contributions in the order of presentation of data in the paper.

The authors declare no conflict of interest.

See the funding table on p. 20.

of prophylactic COVID-19 vaccines of different forms, such as mRNA vaccines (1, 2), adenoviral vectored vaccines (6, 7), inactivated vaccines (3–5), and a recombinant protein vaccines (8), have been authorized for clinical use. Over 13.3 billion doses of these vaccines have been administered worldwide. In spite of these tremendous global efforts and achievements, the continuous emergence of SARS-CoV-2 variants of concern (VOCs) (17–24) with more and more enhanced community transmission and immune evasion poses great challenges to the control of the pandemic, which shows no sign of termination in a short term. So far, SARS-CoV-2 has already caused more than 770 million documented infections and over 6.9 million documented deaths over the world, according to World Health Organization (WHO) online updates (https://www.who.int/emergencies/diseases/novel-coronavirus-2019). The ongoing nature of the COVID-19 pandemic is likely a consequence of multiple factors, such as the presence of a large size of unvaccinated population, which is more than 30% global population (25); the ineffectiveness of the current vaccines to provide sterilizing immunity against SARS-CoV-2 infection (26); and the continuous evolution of the SARS-CoV-2 virus that promotes escape of the virus from vaccine- or natural infection-induced immunity (27, 28).

SARS-CoV-2 is a betacoronavirus that has broad host range (29–33). Receptor usage is a critical determinant for the host range of coronaviruses (32, 34). SARS-CoV-2 utilizes ACE2 as an essential cellular receptor (29, 35–37). It has been found that the SARS-CoV-2 early isolate is able to utilize a wide range of mammalian ACE2 orthologs, but not mouse ACE2, for cell entry *in vitro* (30, 33, 38). Animal studies further show that this virus can infect a number of animal species either through natural infection (e.g., cats, dogs, lions, tigers, mink, and white-tailed deer) or experimental infection (e.g., monkeys, hamsters, ferrets, and tree shrew) (31, 39–43). Among these animal species, cats are known to be highly susceptible to SARS-CoV-2 infection when inoculated intranasally or orally and show prolonged periods of viral shedding (44). It is worth noting that transmission of SARS-CoV-2 from infected animals to humans has also been detected. For example, SARS-CoV-2 outbreaks among minks in farms in several countries have led to transmission of the virus from minks to humans working in those farms (41, 43). In addition, white-tailed deer were reported to be infected with SARS-CoV-2 and spillback to human (42, 43).

The receptor-binding domain (RBD), which locates at residues 330–530 of the viral Spike protein, is responsible for the Spike-ACE2 interaction and binds ACE2 with high affinity (45). The RBD is also the primary target of the neutralizing antibodies elicited by natural infection or vaccination (9–13, 23, 46, 47) and thus a mutation hotspot of immune pressure-driven viral evolution (48–50). The Spike protein is a type I transmembrane protein and assembled as homotrimers on the viral surface. The three RBDs of the Spike trimer form the apex of the viral Spike and sample two distinct conformations: "up" for a receptor-accessible state and "down" for a receptor-inaccessible state (51). The RBD has two subdomains: a core structure and an extended loop, which is named the receptor-binding motif (RBM) and makes most of the contacts with ACE2 (45, 52, 53). As human ACE2 residues on the RBD-binding interface are not fully conserved across mammalian-ACE2 orthologs (30, 51, 53, 54), mutations within the RBD might easily alter cross-species receptor usage by SARS-CoV-2 variants. Indeed, a single amino acid change within the Spike RBD, such as Q498H, Q498Y, or N501Y, is sufficient to confer SARS-CoV-2 ability to utilize mouse ACE2 as entry receptor and establish *in vivo* infection (55–59).

SARS-CoV-2 is a single-stranded RNA virus with moderate mutation and recombination frequencies (27, 28). Mutations have been accumulated in the Spike RBD region of SARS-CoV-2 variants, especially those of the VOCs (17–24). It is possible that some of the mutations might alter these VOCs' binding affinities to animal ACE2 proteins, a key indicator of SARS-CoV-2's host range. In this study, we investigated five WHO-designated VOCs, including Alpha (B.1.1.7) (17, 18), Beta (B.1.351) (19), Gamma (P.1) (20), Delta (B.1.617.2) (21, 22), and Omicron (B.1.1.529) (23, 24), for their Spike protein's interaction with 18 animal ACE2 orthologs.

# RESULTS

## Spike RBDs of the five VOCs have distinct binding affinities to diverse animal ACE2 proteins

We first cloned the Spike genes and the RBD fragments of an early SARS-CoV-2 variant WHU01 (60) and the five VOCs (Fig. 1A through C). All the VOCs except Delta carry an

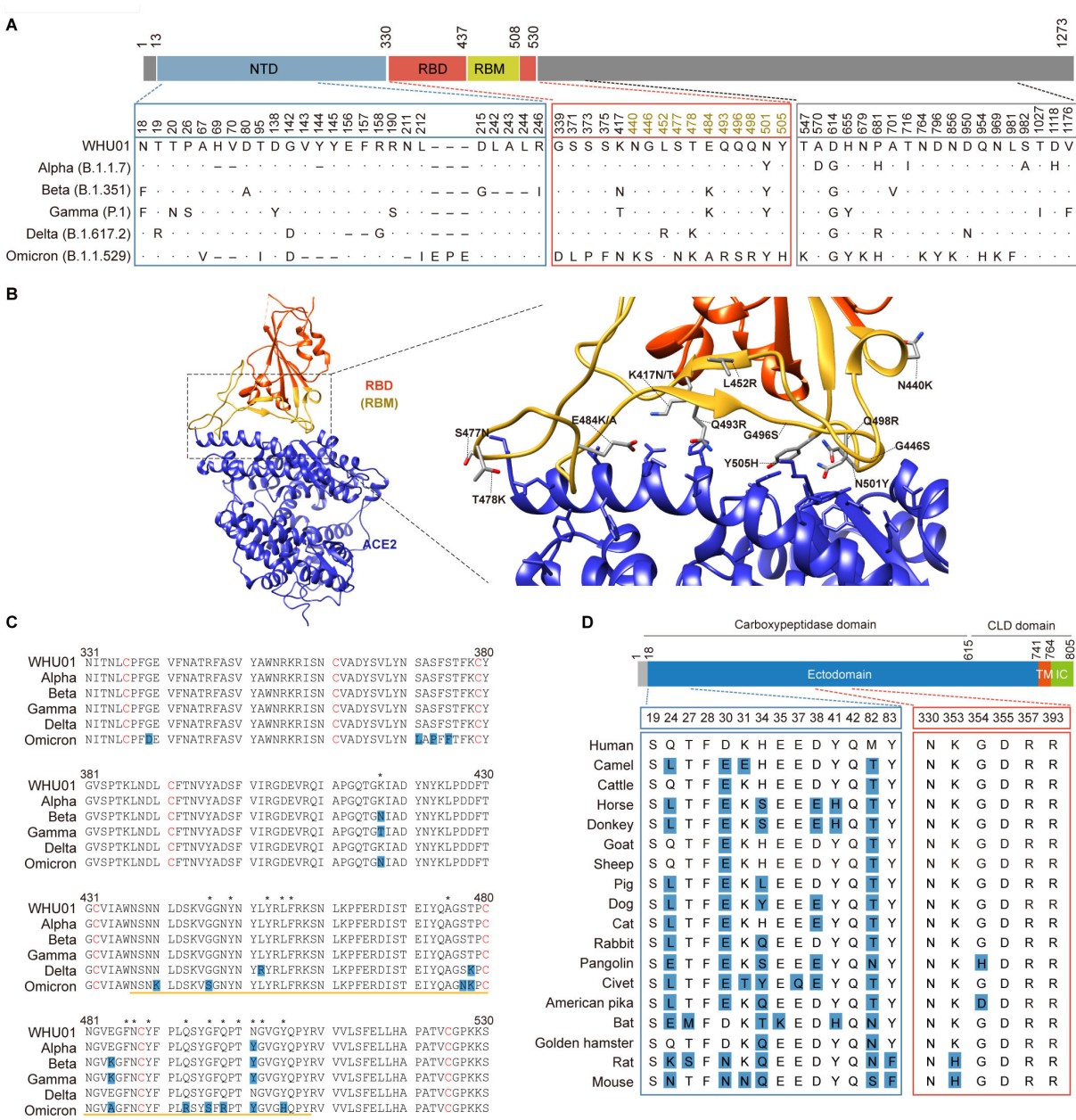

**FIG 1** SARS-CoV-2 Spike variants and ACE2 orthologs investigated in the following studies. (A) The Spike genes of SARS-CoV-2 (WHU01) and five VOCs investigated in this study are illustrated. The RBD is in red, and the RBM is in goldenrod. Spike mutations associated with each variant are indicated. (B) Interaction between the SARS-CoV-2 WHU01 RBD (red) and human ACE2 (blue) involve a large number of contact residues (PDB accession no. 6M0J). The RBM within the RBD is indicated in goldenrod. The human ACE2 residues within 4 Å from RBD atoms are shown. All SARS-CoV-2 variant-associated RBM mutations investigated in this study are shown and labeled. (C) Amino acid sequences of the SARS-CoV-2 WHU01 and five VOC RBDs are aligned, with residues different from the corresponding ones in WHU01 highlighted in blue. The stars indicate RBD residues within 4 Å from ACE2 atoms. The yellow lines indicate the RBM region. (D) Amino acid sequences of the indicated 18 ACE2 orthologs are aligned, with only residues within 4 Å from RBD atoms shown here. The numbering is based on human ACE2 protein, and the residues different from the corresponding ones in human ACE2 are highlighted in blue.

N501Y signature mutation in the RBD region. In addition, Beta, Gamma, and Omicron have mutations at two same RBD residues, K17 and E484. Omicron then has 12 additional mutations in the RBD region. Delta, in contrast, only has two distinct mutations, L452R and T478K, in the RBD region. In addition to the mutations in the RBD regions, the five VOCs also carry a number of additional lineage-defining amino acid substitutions in the remaining region of the corresponding Spike proteins (17–24). We then constructed expression plasmids for 18 animal ACE2 proteins (Fig. 1D; Fig. S1A and B; Table S1), each with an S-tag at the N-terminus, and expressed the six RBDs as human IgG1 Fc fusion proteins (RBD-huFc; Fig S1C). Purified RBD-huFc proteins were then used to perform surface staining of 293T cells transfected with each of the 18 S-tagged ACE2 orthologs or a vector plasmid control. Based on the intensity of the S-tag staining signal in flow cytometry analysis, ACE2-positive 293T cells were gated into two populations, ACE2-high [mean fluorescence intensity (MFI): $2.5 \times 10^3$ to $2.5 \times 10^4$] and ACE2-low (MFI: $2.5 \times 10^2$ to $2.5 \times 10^3$; Fig. S2). RBD binding was analyzed independently in the two ACE2-positive cell populations (Fig. 2A and B; Fig. S2). Interestingly, while Delta RBD showed a pattern of animal ACE2 binding affinities almost identical to that of the WHU01 RBD, RBDs of the other four VOCs (Alpha, Beta, Gamma, and Omicron) showed gain of increasingly high affinity to mouse ACE2 while loss of affinity to increasing number of animal ACE2 orthologs (Fig. 2C and D). Specifically, Alpha RBD showed gain of binding to mouse ACE2 protein, markedly increased binding to ACE2 orthologs of multiples species (e.g., cattle and pig), and decreased binding to ACE2 orthologs of horse, donkey, and *Rhinolophus sinicus* bat. In the case of Beta and Gamma RBDs which have similar signature mutations, almost identical binding patterns were observed. Similar to Alpha RBD, Beta and Gamma RBDs also showed gain of binding to mouse ACE2 protein. In addition, Beta and Gamma RBDs showed weakened binding to dog and pangolin ACE2 proteins and complete loss of binding to horse, donkey, and *Rhinolophus sinicus* bat ACE2 orthologs. More interestingly, although Omicron RBD exhibited further enhanced binding to mouse ACE2, it showed markedly decreased or complete loss of binding to ACE2 ortholog of eight animal species, including that of horse, donkey, pig, dog, cat, pangolin, American pika, and *Rhinolophus sinicus* bat (Fig. 2C and D). In the following sections, we dissect each VOC's significant binding affinity changes one by one through more detailed studies. As the interactions observed in the ACE2-low cells are more likely to be high-affinity interactions and thus more likely to be physiologically relevant, we opt to focus more on the interactions or changes observed in the ACE2-low cells in the following studies.

## Alpha (B.1.1.7) RBD has markedly enhanced affinity to cattle and pig ACE2 proteins

It's well established that Alpha RBD has a higher binding affinity to human ACE2 receptor and Alpha variant has higher community transmissibility than the SARS-CoV-2 WHU01 strain (61–66). Structural studies show that the Y501 residue of Alpha RBD inserts into a cavity at the binding interface and forms a perpendicular π–π stacking interaction with the Y41 residue of ACE2 (62, 63). Perhaps because of a same mechanism, we observed that Alpha RBD binds more efficiently than the WHU01 RBD does to most of the tested ACE2 orthologs, except for the orthologs of horse, donkey, *Rhinolophus sinicus* bat (isolate Rs-3357), pangolin, and civet, the first three of which have an H41 rather than a Y41 (Fig. 1D, 2D, and 3A). Since cattle and pigs are extremely important livestock animals which serve as two major sources of meat for humans, we further quantitatively measured interaction kinetics of Alpha RBD binding to cattle and pig ACE2 proteins. Cat was reported to be very susceptible to SARS-CoV-2 infection (31, 40); we, therefore, also included cat ACE2 in the analysis. We first expressed recombinant RBD proteins of the indicated variants as monomeric proteins and the extracellular domains of the indicated ACE2 orthologs as human IgG1 Fc fusion proteins (ACE2-huFc; Fig. S3A and B). Purified ACE2-huFc and RBD proteins were then used as immobilized receptors and analytes, respectively, in a surface plasmon resonance (SPR) assay and a biolayer interferometry (BLI)-based assay (Fig. 3B; Fig. S3C and D; Table S2). Compared to the

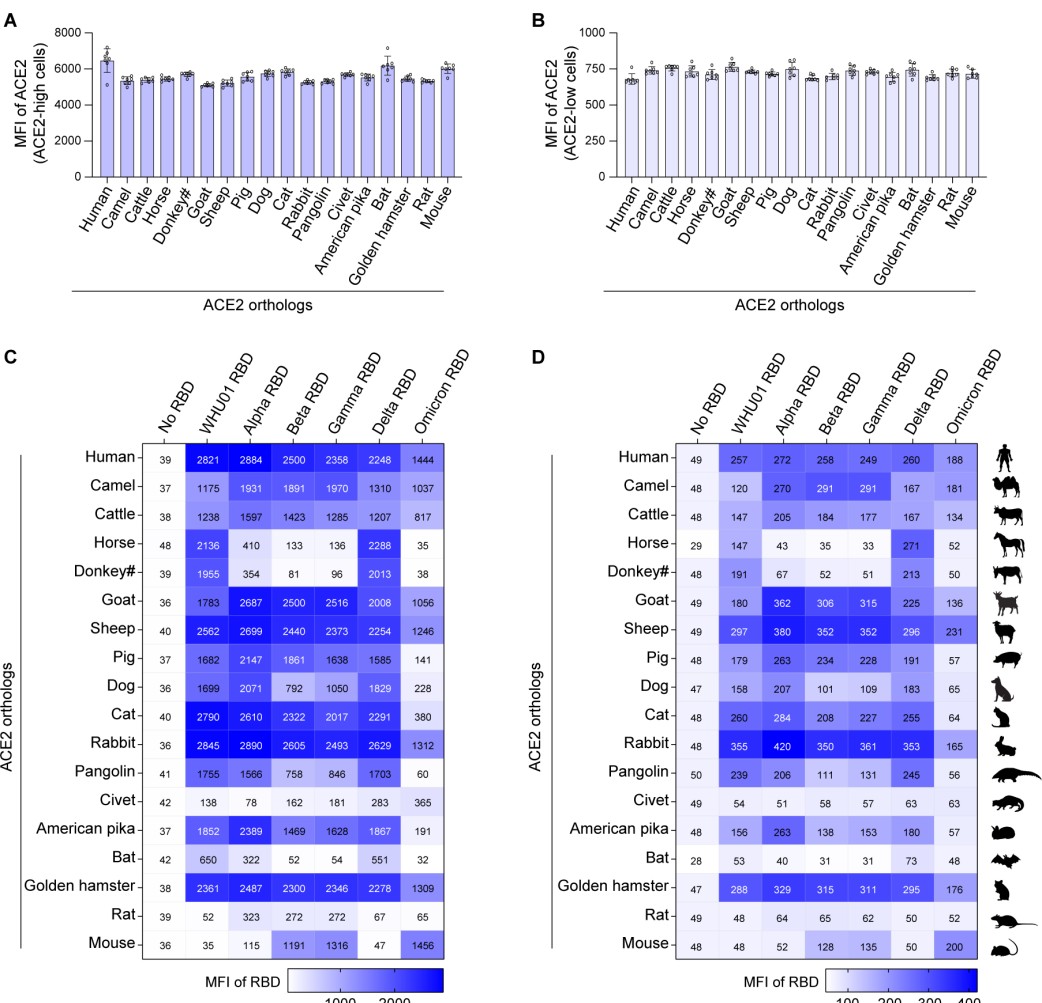

**FIG 2** Five VOC RBDs have distinct binding affinities to 18 animal ACE2 orthologs. 293T cells expressing 1 of the 18 animal ACE2 orthologs were double stained for ACE2 protein expression as well as binding by IgG-Fc-fused dimeric RBD proteins of the indicated SARS-CoV-2 variants. Cells were then analyzed using flow cytometry. ACE2-positive cells were first gated into ACE2-high (A) and ACE2-low (B) populations based on the MFI of ACE2 staining signals. ACE2-low cells have ACE2 MFI in the range of $2.5 \times 10^3$ to $2.5 \times 10^4$, while ACE2-high cells have ACE2 MFI in the range of $2.5 \times 10^2$ to $2.5 \times 10^3$. The dot plot raw data and gating strategy for the entire flow cytometry experiments are presented in Fig. S2. RBD-binding signals in ACE2-high (C) and ACE2-low (D) populations were separately analyzed, and MFI values for each RBD/ACE2 interaction are shown in color-coded heatmap grids. The dot plot raw data for the entire flow cytometry experiments are presented in Fig. S2. Data shown are representative of two independent experiments with similar results, and data points in panels A and B represent mean ± SD of seven biological replicates.

human ACE2 and WHU01 RBD interaction, cattle and pig ACE2 proteins, respectively, showed a 5.4- and 8.8-fold lower affinity to WHU01 RBD. However, the N501Y mutation in Alpha RBD increased RBD affinity to cattle and pig ACE2 proteins by 14.7- and 13.5-fold, respectively, resulting in the fact that the affinities of cattle ($K_D$ = 47.6 nM) and pig ($K_D$ = 84.4 nM) ACE2 proteins to Alpha RBD are higher than the affinity of human ($K_D$ = 129 nM) ACE2 to WHU01 RBD (Fig. 3B; Table S2). In contrast, the N501Y mutation in Alpha RBD only resulted in a 5.0- and 2.6-fold increased affinity to human and cat ACE2 proteins, respectively (Fig. 3B; Table S2). Consistent with the binding kinetics data, recombinant cattle and pig ACE2-huFc proteins neutralized Alpha pseudovirus ~17- and ~80-fold more efficient than neutralizing WHU01 pseudovirus, respectively, while recombinant human and cat ACE2-huFc proteins only showed 4- and 2.5-fold differences (Fig. 3C). To understand the molecular basis of cattle and pig ACE2 proteins' more

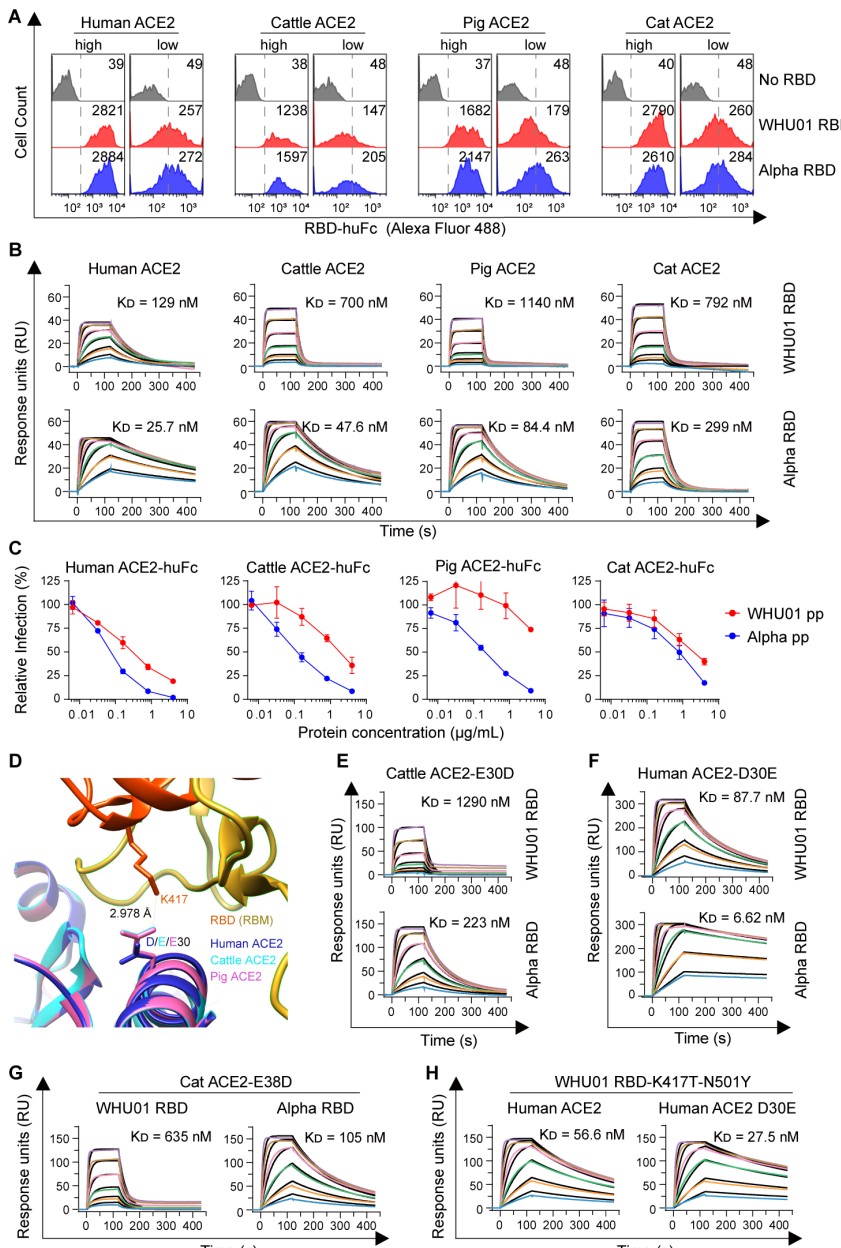

**FIG 3** Alpha RBD has markedly enhanced affinity to cattle and pig ACE2 proteins. (A) Flow cytometry histogram data obtained from the experiments shown in Fig. 2 are shown for the indicated RBD variants binding to cell surface-expressed animal ACE2 orthologs. RBD-binding MFI values for ACE2-high and ACE2-low cells are separately shown in each figure. (B) SPR measurements of interaction kinetics were performed using one of the indicated animal ACE2 ectodomain dimers as immobilized ligand and one of the indicated monomeric RBD proteins at 2,000 nM, 1,000 nM, 500 nM, 250 nM, 125 nM, and 62.5 nM as analytes. The raw curves are shown in colors, and the fitted curves obtained from a 1:1 binding model are presented in black. The full binding kinetics data are provided in Table S2. (C) Human IgG1 Fc fusion proteins of the indicated animal ACE2 ectodomains (ACE2-huFc) were tested for their neutralization activity against WHU01 or Alpha Spike-pseudotyped reporter viral particle (pp) infection of an HeLa-hACE2 stable cell line. Luciferase reporter expression at 48 hours post pseudovirus infection was presented as percentage to the reporter expression in the absence of an animal ACE2-huFc protein. (D) Structure models of cattle and pig ACE2 proteins were first generated through SWISS-MODEL, using a cryo-EM structure of human ACE2 ectodomain and SARS-CoV-2 N501Y mutant RBD complex (PDB

**FIG 3** (Continued)

accession no. 7MJN) as a modeling template. Modeled cattle ACE2 (cyan) and pig ACE2 (pink) structures were then superimposed to the Cryo-EM structure (PDB accession no. 7MJN) of human-ACE2 (blue) in complex with N501Y-RBD (red and goldenrod) with UCSF Chimera. The RBD-K417 (red) and ACE2-D30 (blue) or ACE2-E30 (cyan and pink) residues that can form a salt bridge interaction are shown and labeled. (E–H) SPR experiments similar to panel B, except that the analytes and immobilized ligands used here are the indicated monomeric RBD variants and dimeric animal ACE2 mutant proteins, respectively. The full binding kinetics data are provided in Table S2. Data shown in this figure are representative of two or three independent experiments performed by two different people with similar results, and data points in panel C represent mean ± SD of three biological replicates.

pronounced response to RBD N501Y mutation, mutagenesis studies were performed. Since on the ACE2-RBD binding interface, compared to human ACE2, cattle and pig ACE2 proteins have a common D30E substitution that might form a stable salt bridge with interaction with RBD's K417 residue (38, 53, 67, 68) (Fig. 1C, D and 3D), we generated a D30E mutant of human ACE2-huFc and a E30D mutant of cattle ACE2-huFc for SPR analysis. Interestingly, the E30D mutation in cattle ACE2 resulted in the affinity difference between WHU01 and Alpha (N501Y) RBDs decreased from the original 14.7-fold (E30) to 5.8-fold (D30; Fig. 3B and E). On the other hand, the D30E mutation in human ACE2 resulted in the affinity difference between WHU01 and Alpha RBDs increased from the original 5.0-fold (D30) to 13.2-fold (E30; Fig. 3B and F). Although cat ACE2 also has an E30, it has an additional D38E substitution compared to human ACE2 (Fig. 1D). An E38D mutation in cat ACE2 makes it more sensitive to the N501Y mutation in the RBD (Fig. 3B and G), consistent with the data from our previous molecule dynamic study (69). These data consistently indicate that the D30E substitution from human to cattle ACE2 proteins results in cattle ACE2's more pronounced response to RBD N501Y mutation. Interestingly, when a K417T mutation was introduced to Alpha RBD, the D30E mutation in human ACE2 could still produce a 2.1-fold affinity increase (Fig. 3H), suggesting that, in addition to the negatively charged head group, the hydrophobic side chain of the Glu30 residue also contributes to cattle ACE2's high-affinity binding to Alpha RBD.

## Beta (B.1.351) and Gamma (P.1) Spikes have lost binding to ACE2 orthologs of horse and *Rhinolophus sinicus* bat

Compared to WHU01 RBD, Alpha RBD that only carries an N501Y substitution showed decreased staining of 293T cells expressing *Rhinolophus sinicus* bat or horse ACE2 (Fig. 2C and 4A). Beta and Gamma RBDs, both of which also carry the N501Y substitution and two more mutations (K417N/T and E484K), exhibited more pronounced decrease in staining of these cells and complete loss of binding in the ACE2-low cell population (Fig. 2C and 4A). Similarly, Alpha, Beta, and Gamma RBDs also displayed decreased or loss of binding to 293T cells expressing donkey ACE2, which has residues on the RBD contact interface identical to that of horse ACE2 (Fig. 1D and 2B; Fig. S1). We then focused on the interaction changes observed with horse and bat ACE2 orthologs. We first performed interaction kinetics analysis and validated the above interaction changes observed with bat and horse ACE2 orthologs (Fig. 4B; Fig. S4A and B; Table S3). To understand the impact of each mutation on RBD binding to bat or horse ACE2, we then performed mutagenesis studies on RBD. Cell surface staining experiments comparing WHU01 RBD and its mutant proteins, each carrying a single amino-acid mutation from the Alpha, Beta, and Gamma variants, showed that all the tested mutations (K417N/T, E484K, and N501Y) decreased RBD binding to bat ACE2 and that the E484K mutation produced the most pronounced effect (Fig. 4C; Fig. S4C and D). This is likely because that the K417N/T mutation impairs an important salt bridge interaction between RBD K417 and ACE2 D30 residues; then, an RBD K484 residue could introduce a repulsion against bat ACE2 K35 residue, while an RBD-Y501 residue might compete with the aliphatic part of ACE2-K353 residue for interacting with the aromatic ring of RBD-Y505 residue (70) (Fig. 4D). In the case of horse ACE2, likely because of similar mechanisms,

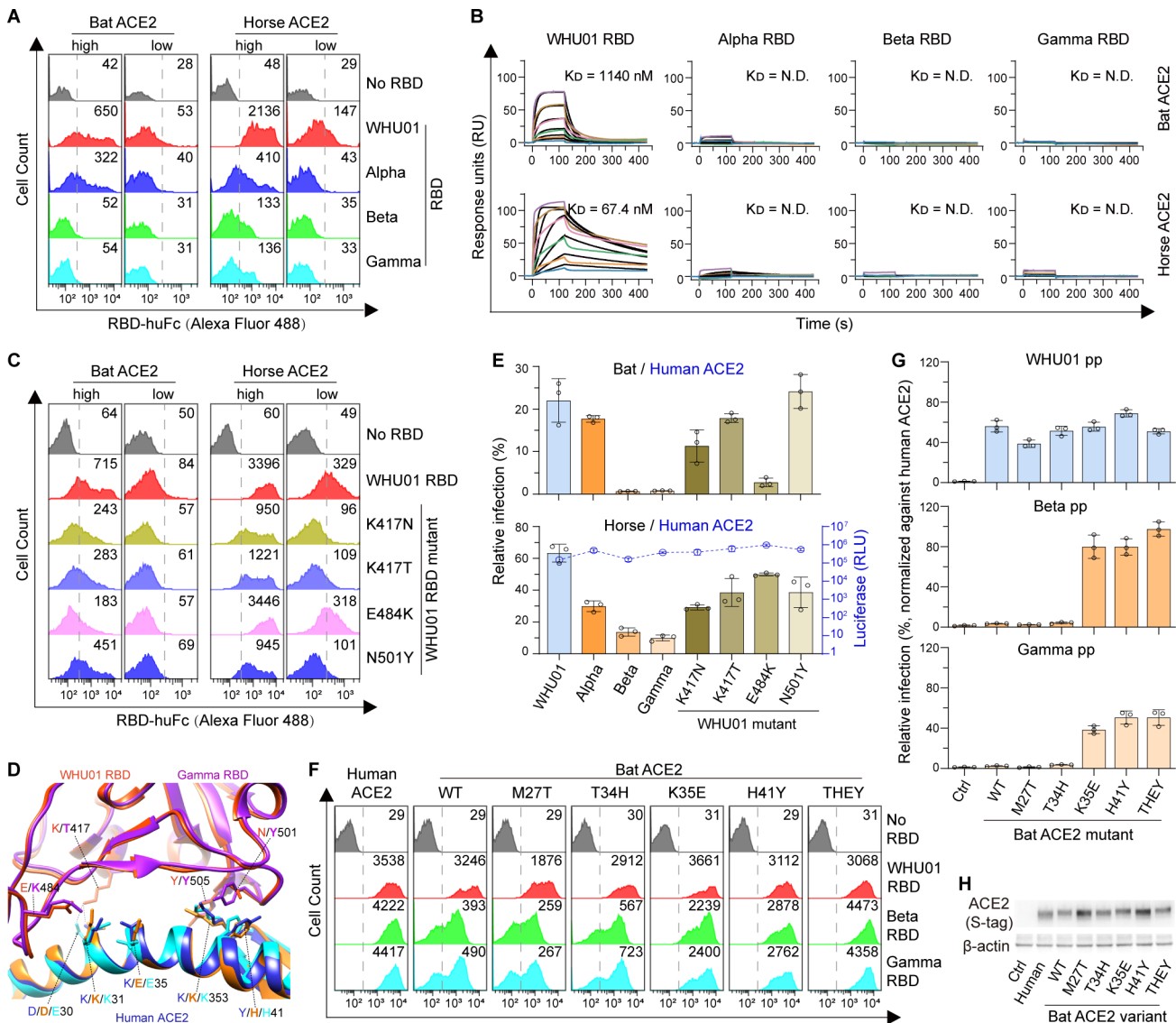

**FIG 4** Beta and Gamma Spikes have lost binding to *Rhinolophus sinicus* bat and horse ACE2 orthologs. (A) Flow cytometry histogram data obtained from the experiments shown in Fig. 2 are shown for the indicated RBD variants binding to cell surface-expressed *Rhinolophus sinicus* bat or horse ACE2 orthologs. (B) SPR measurements of interaction kinetics were performed using dimeric *Rhinolophus sinicus* bat or horse ACE2 ectodomain as immobilized ligand and one of the indicated monomeric RBD proteins as analyte. The full binding kinetics data are provided in Table S3. (C) Flow cytometry detection of interactions between the indicated RBD dimers and cell surface-expressed *Rhinolophus sinicus* bat or horse ACE2 proteins. (D) A structure model of *Rhinolophus sinicus* bat ACE2 protein was first generated using a crystal structure of human ACE2 ectodomain in complex with the gamma RBD (PDB ID: 7NXC) as a modeling template. Modeled bat ACE2 structure, together with a human-ACE2/SARS-CoV-2-RBD complex structure (PDB accession no. 6MOJ), and a horse-ACE2 structure (PDB accession no. 7W6U) were superimposed to the human-ACE2/Gamma-RBD complex structure (PDB accession no. 7NXC). Orange red and purple indicate the WHU01 RBD and the Gamma RBD, respectively. Blue, orange, and cyan indicate human, bat, and horse ACE2 proteins, respectively. Key ACE2 residues that may be affected by the Gamma signature mutations (K417T, E484K, and N501Y) are shown and labeled. (E) SARS-CoV-2 pseudovirus infection experiments using the indicated SARS-CoV-2 variant Spike-pseudotyped reporter viruses for infection and 293T cells expressing *Rhinolophus sinicus* bat, horse, or human ACE2 as the target cells. Infection signals of each pseudovirus supported by bat or horse ACE2 (bar graphs) were calculated as percentage of infection relative to infection signals of corresponding pseudovirus supported by human ACE2. The original luciferase signals supported by human ACE2 (blue curve) are plotted on the right Y-axis, showing that all the pseudoviruses maintain good infectivity to human ACE2-expressing cells. (F) Flow cytometry detection of interactions between the indicated RBD dimers and cell surface-expressed human ACE2 or *Rhinolophus sinicus* bat ACE2 mutants. WT indicates bat ACE2 wild-type sequence. THEY indicates a bat ACE2 mutant that simultaneously carries the M27T, T34H, K35E, and H41Y mutations. (G) SARS-CoV-2 pseudovirus infection experiments similar to panel E, except that the indicated pseudoviral particles (pp) and 293T cells expressing the indicated *Rhinolophus sinicus* bat ACE2 mutants or human ACE2 were used here. Infection signals of each pseudovirus supported by bat ACE2 mutants were calculated as percentage of infection relative to signals of

**FIG 4** (Continued)

corresponding pseudovirus supported by human ACE2. (H) Western blot detection of ACE2 expression in the transfected 293T cells used in panel G. Data shown are representative of two or three independent experiments performed by two different people with similar results, and data points in panels E and G represent mean ± SD of three biological replicates.

the K417N/T and N501Y RBD mutations also substantially decreased binding. In contrast, the E484K RBD mutation has no impact on binding to horse ACE2, consistent with the fact that horse ACE2 has a negatively charged E35 residue rather than a positively charged K35 residue (Fig. 4C and D). The above molecule interaction-based findings were then further validated in the ACE2-mediated pseudovirus infection assay (Fig. 4E; Fig. S4E). To further investigate molecular bases of bat ACE2's complete loss of interaction with Beta and Gamma, we then performed mutagenesis studies on bat ACE2 protein. Compared to human ACE2, *Rhinolophus sinicus* bat ACE2 has six amino acid substitutions at the RBD-binding interface. Since human ACE2 residues at four of these positions are either relatively conserved among other ACE2 orthologs or reside in close proximity to the mutated RBD residues in ACE2-RBD complex (Fig. 1), we generated five bat ACE2 mutants. Each one of the first four mutants carries a bat-to-human single amino acid mutation, M27T, T34H, K35E, or H41Y. The fifth one, THEY, is a mutant that carries all the four mutations (M27T-T34H-K35E-H41Y). Cell surface staining and pseudovirus infection assays consistently showed that both K35E and H41Y mutations enabled bat ACE2 to efficiently interact with Beta and Gamma Spikes (Fig. 4F through H; Fig. S4F). The bat ACE2 K35E mutation, which might compensate the E484K RBD mutation's repulsion effect (Fig. 4D), offsetted ~50% of loss of Spike interaction caused by Beta or Gamma RBD mutations (Fig. 4F; Fig. S4F). The H41Y mutation offset ~60% of Beta or Gamma RBD mutation's effect (Fig. 4F and F), possibly through compensating the N501Y RBD mutation's negative impact (Fig. 4D). The THEY mutant that carries both K35E and H41Y mutations fully offset Beta or Gamma RBD mutation's effect.

## Omicron (B.1.1.529) RBD mutations are optimal for electrostatic interactions with mouse ACE2

We have previously found that, in contrast to the WHU01 strain, VOCs Alpha, Beta, and Gamma variants are able to use mouse ACE2 as entry receptor (71). Consistent with this finding, an *in vivo* study reported that the VOCs Alpha, Beta, and Gamma were all able to infect mouse, but only the Beta and Gamma variants could replicate to high titers in the lungs (58). Here, we found that, compared to Beta and Gamma variants, Omicron variant has gained further enhanced affinity to mouse ACE2 (Fig. 5A through C). Using SPR assay, we found that Omicron RBD binds mouse ACE2 with a strong affinity of ~230 nM, which is around 10-fold stronger than the affinity of Beta or Gamma RBD to mouse ACE2 and in the same range as the affinity of WHU01 RBD to human ACE2 (Fig. 5B; Table S4). Consistent with these binding data, pseudovirus infection assay showed that Omicron variant utilized cell surface mouse ACE2 for entry more efficiently than Beta or Gamma variants did (Fig. 5C). When soluble mouse ACE2 protein was tested in a pseudovirus neutralization assay, it neutralized Omicron pseudovirus infection more potently than Beta or Gamma pseudovirus (Fig. 5D). Then, we performed RBD mutagenesis studies and found that seven RBD mutations (K417N, T478K, E484A, Q493R, G496S, Q498R, and N501Y) could facilitate RBD binding to mouse ACE2 (Fig. 5E; Fig. S8 and S5A). Three of these RBD mutations (K417N, T478K, and E484A) shown detectable but minimal contribution (Fig. S5A). The other four (Q493R, G496S, Q498R, and N501Y) showed much more significant enhancement on binding to mouse ACE2, with Q493R and N501Y showed the most pronounced effects (Fig. 5E through H; Fig. S5; Table S4).

To gain more insights into the interactions between Omicron RBD and mouse ACE2, we built models for WHU01-RBD:mACE2 and Omicron-RBD:mACE2 complexes respectively, and each performed triplicate molecular dynamics (MD) simulations to understand how Omicron-RBD key residues contribute to the binding with mouse ACE2. From the MD trajectory of the Omicron-RBD:mACE2 complex, we found that RBD R493 residue not

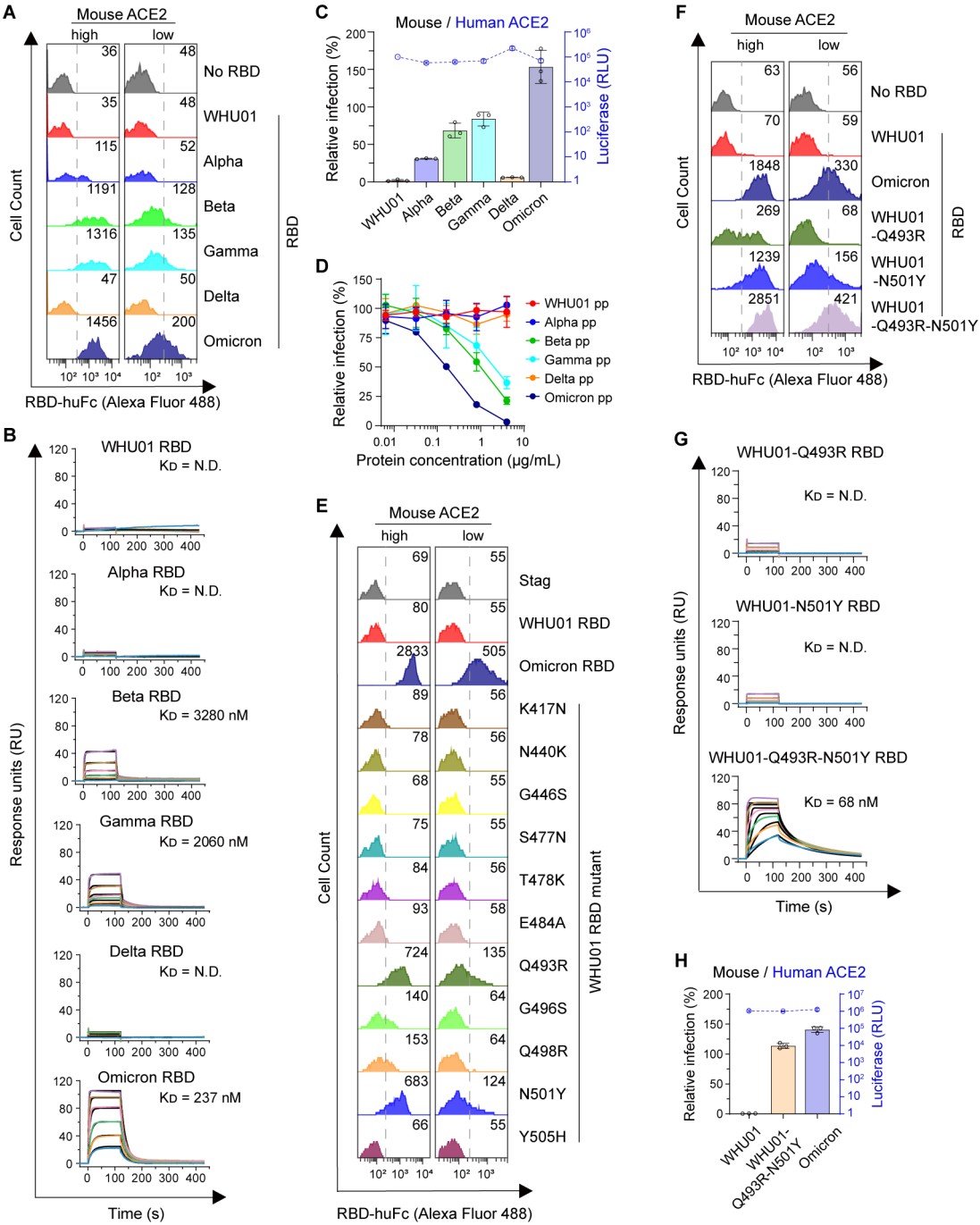

**FIG 5** Omicron has gained high-affinity binding to mouse ACE2 mainly through the RBD Q493R, G496S, Q498R, and N501Y mutations. (A) Flow cytometry histogram data obtained from the experiments shown in Fig. 2 are shown for the indicated RBD variants binding to cell surface-expressed mouse ACE2. (B) SPR measurements of interaction kinetics were performed using dimeric mouse ACE2 ectodomain as immobilized ligand and one of the indicated monomeric RBD proteins as analyte. The full binding kinetics data are provided in Table S4. (C) SARS-CoV-2 pseudovirus infection experiments similar to Fig. 4E using the indicated SARS-CoV-2 variant Spike-pseudotyped reporter viruses for infection and 293T cells expressing mouse or human ACE2 as the target cells. Infection signals of each pseudovirus supported by mouse ACE2 (bar graph) were calculated as percentage of infection relative to infection signals of corresponding pseudovirus supported by human ACE2. The original luciferase signals supported by human ACE2 (blue curve) are plotted on the right Y-axis, showing that all the pseudoviruses maintain good infectivity to human ACE2-expressing cells. (D) SARS-CoV-2 pseudovirus neutralization experiments similar to Fig. 3C using mouse ACE2-huFc as the neutralizing agent and the indicated SARS-CoV-2 variants pseudoviral particles (pp) for infection of the HeLa-hACE2 stable cell line. (E) Flow cytometry histogram data obtained from the experiments shown in Fig. 7B are shown for the indicated RBD variants binding to cell surface-expressed mouse ACE2. (F) Flow cytometry detection of interactions between the indicated RBD dimers and cell surface-expressed mouse ACE2 protein. (G) SPR

**FIG 5** (Continued)

measurements of interaction kinetics were performed using dimeric mouse ACE2 ectodomain as immobilized ligand and one of the indicated monomeric RBD proteins as analyte. The full binding kinetics data are provided in Table S4. (H) Pseudovirus infection experiments similar to C using the indicated SARS-CoV-2 variant Spike-pseudotyped reporter viruses. Data shown are representative of two or three independent experiments performed by two different people with similar results, and data points in panels C, D, and H represent mean ± SD of three biological replicates.

only forms a salt bridge with mouse ACE2 E35 residue but also donates a hydrogen bond to mouse ACE2 N31 residue (Fig. 6A through D). Meanwhile, RBD R498 and S496 residues each donate a hydrogen bond to mouse ACE2 D38 and H353 residues, respectively, which further strengthens the hydrogen bond between RBD Y449 residue and mouse ACE2 D38 residue (Fig. 6A through D). We then performed molecular mechanics with generalized Born and surface area solvation (MM/GBSA) analysis of RBD in complex with human or mouse ACE2 to understand the energetic contribution of key interface

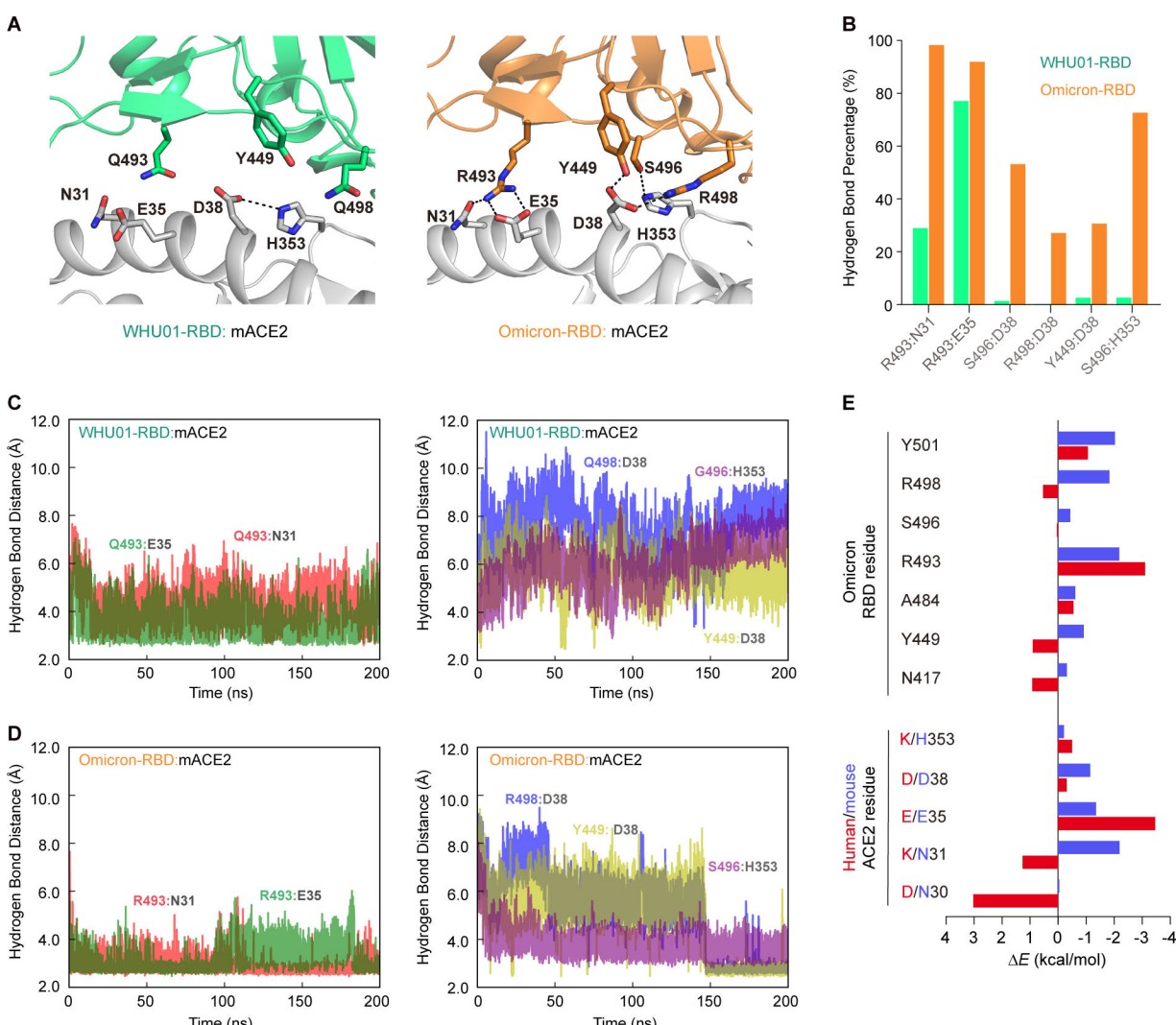

**FIG 6** Omicron RBD mutations are optimal for electrostatic interactions with mouse ACE2. (A) A homology model of Omicron-RBD:mACE2 was built, and triplicate MD simulations were performed to understand how RBD R493, S496, and R498 promote the affinity with mouse ACE2. From the MD trajectory of the Omicron-RBD:mACE2 complex, RBD R493 forms a salt bridge and a hydrogen bond with mouse ACE2 E35 and N31, respectively. RBD R498 and S496 each form a hydrogen bond with mouse ACE2 D38 and H353, respectively, which further strengthens the hydrogen bond between RBD Y449 and mouse ACE2 D38. (B) Percentages of the key hydrogen bond/salt bridge at the interface. (C and D) Time series of key hydrogen bond in WHU01-RBD:mACE2 complex (C) and Omicron-RBD:mACE2 complex (D). (E) MM/GBSA analysis result showing change of residue-specific interaction energy from WHU01 RBD to Omicron RBD upon binding mouse ACE2 (blue) or human ACE2 (red).

residues to the total RBD/ACE2 binding. Unexpectedly, all the calculable Omicron RBD key residues (N417, Y449, A484, R493, S496, R498, and Y501) and mouse ACE2 residues (N31, E35, D38, and H353) were found favorable to the RBD/ACE2 interaction, with the interactions between RBD R493 residue and mouse ACE2 N31 and E35 residues showing the largest contribution (Fig. 6E), which is fully consistent with the flow cytometry and SPR data (Fig. 5E through G). In contrast, Omicron RBD residues N417, Y449, and R498 and human ACE2 residues D30 as well as K31 were found negatively impacting the RBD/ACE2 interaction (Fig. 6E) (72). These data indicate that the Omicron RBD mutations are optimal for electrostatic interactions with mouse ACE2 and suboptimal for interactions with human ACE2.

## Omicron (B.1.1.529) has markedly decreased affinity to multiple tested ACE2 orthologs

Although Omicron RBD has gained further enhanced affinity to mouse ACE2 (Fig. 5), it has markedly decreased affinity to multiple tested ACE2 orthologs, including that of some livestock animals (horse and pig), pet animals (dog and cat), and wild animals (pangolin and *Rhinolophus sinicus* bat; Fig. 2; Fig. S6A). These binding changes were further validated with SPR interaction kinetics analysis (Fig. 7A; Table S5) and pseudovirus infection assays (Fig. S6B). To understand the impact of each Omicron mutation on RBD binding to each ACE2 orthologs, we performed mutagenesis studies on WHU01 RBD. Cell surface staining and SPR experiments were used to compare WHU01 RBD and its mutant proteins, each carrying amino acid mutations from the Omicron variant. Interestingly, although the Q493R and G496S mutations could facilitate RBD binding to mouse ACE2 (Fig. S5A), they were found detrimental to the RBD interaction with human ACE2 (Fig. 7B and C), consistent with the MD simulation data (Fig. 6). The K417N, G496S, and Y505H substitutions were identified as three major contributors that commonly have negative impact on RBD binding to these eight ACE2 orthologs (Fig. 7B; Fig. S6D, S7, and S8). Consistently, when these three mutations (K417N, G496S, and Y505H) were simultaneously introduced to WHU01 RBD (WHU01 RBD-NSH), RBD binding to these eight ACE2 orthologs was abolished (Fig. 7D; Fig. S9). Since both K417N and Y505H mutations are present in Omicron's major sublineages (47, 73, 74), the above data also suggest that, in contrast to the WHU01 and Delta variants, the Omicron lineage might be generally less hazardous to livestock and pet animals.

## DISCUSSION

In this study, we investigated an early SARS-CoV-2 isolate and five VOCs (Alpha, Beta, Gamma, Delta, and Omicron) for their Spike protein's interaction with 18 animal ACE2 orthologs. Similar studies involving a panel of animal ACE2 orthologs have previously only been performed for the early isolate Spike (30, 31, 33, 38). Since receptor usage is a critical determinant for the host range of coronaviruses (32), the current study concerning 108 distinct interactions (18 animal ACE2 orthologs × 6 major SARS-CoV-2 Spike variants) provides more comprehensive and useful information for understanding possible cross-species transmission risks of all the five VOCs.

Here, we found that Alpha RBD binds more efficiently than the WHU01 RBD does to multiple ACE2 orthologs, including that of cattle and pig (Fig. 3). Multiple previous studies have shown that cattle and pigs are not or only weakly susceptible to experimental infection with SARS-CoV-2 (31, 75–78). Consistent with this, here, we found that both cattle and pig ACE2 proteins have significantly (5.4- and 8.8-fold) lower affinities than that of human ACE2 to the RBD of an early isolate WHU01. However, in the case of the Alpha RBD, we observed that its N501Y signature mutation increased RBD affinity to cattle and pig ACE2 proteins by 14.7- and 13.5-fold, respectively, resulting in the affinities of cattle and pig ACE2 proteins to Alpha RBD ($K_D = 47.6$ nM for cattle and 84.4 nM for pig) being even higher than the affinity of human ACE2 to WHU01 RBD ($K_D = 129$ nM). In addition, recent structural studies have shown that Spike trimers of multiple SARS-CoV-2

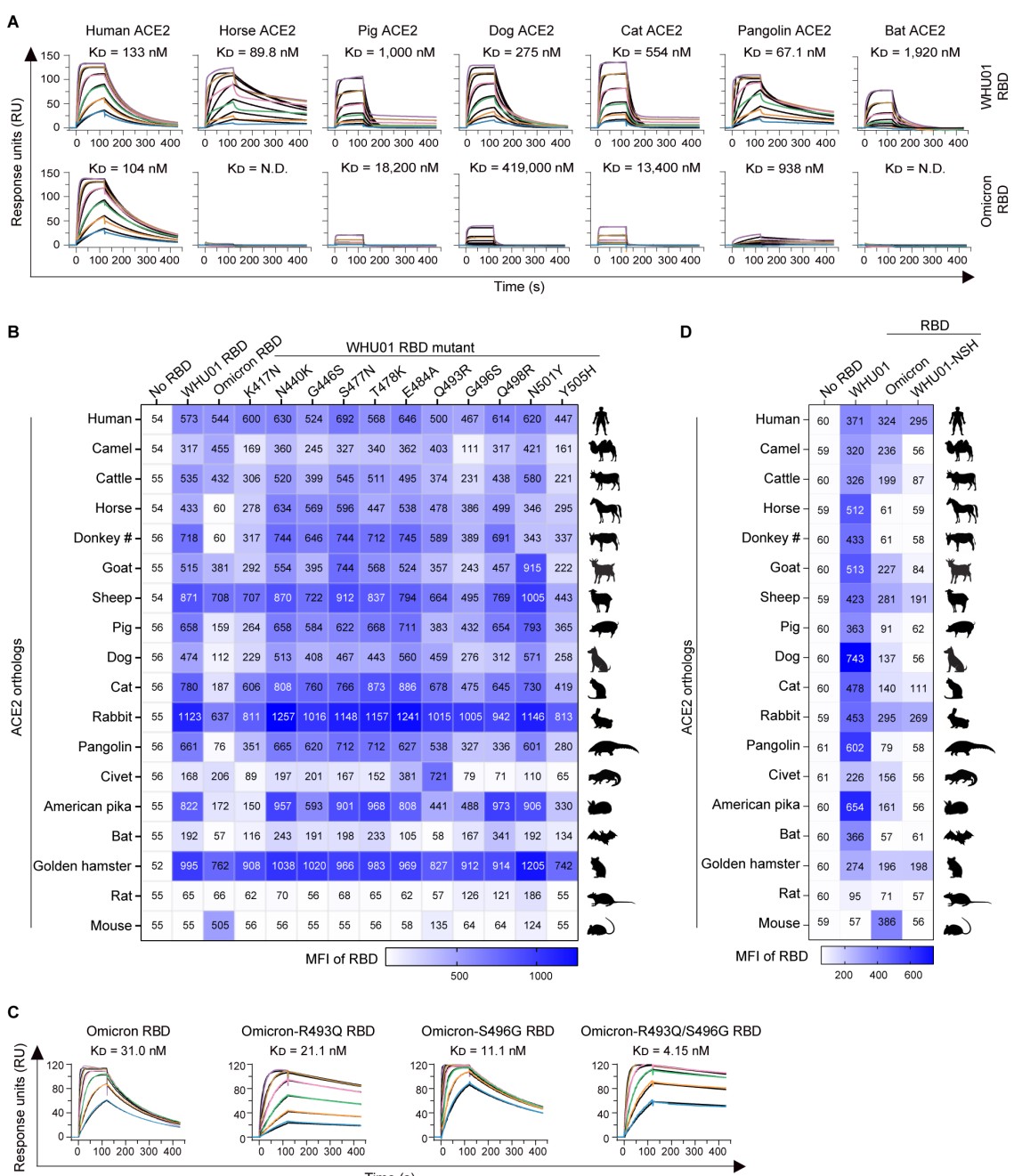

FIG 7 Omicron has lost affinity to multiple ACE2 orthologs. (A) SPR measurements of interaction kinetics were performed using one of the indicated animal ACE2 ectodomain dimers as immobilized ligand and monomeric WHU01 or Omicron RBD as analyte. The full binding kinetics data are provided in Table S5. (B) Flow cytometry detection of interactions between the indicated RBD dimers and cell surface-expressed animal ACE2 orthologs. RBD binding signals in ACE2-high and ACE2-low populations were separately analyzed, with the ACE2-low data shown here and the ACE2-high data shown in Fig. S8. MFI values for each RBD/ACE2 interaction are shown in color-coded heatmap grids. The dot plot raw data for the entire flow cytometry experiments are presented in Fig. S7. (C) SPR experiments similar to A except that the human ACE2 proteins were used as immobilized ligand and the indicated Omicron RBD proteins were used as analyte. (D) Flow cytometry experiments similar to B except that the indicated RBD dimers were used for cell surface ACE2 staining. WHU01-NSH indicates a WHU01 RBD mutant that simultaneously carries K417N, G496S, and Y505H mutations. The dot plot raw data for the entire flow cytometry experiments are presented in Fig. S9A. MFI values for each RBD/ACE2 interaction are shown in color-coded heatmap grids, with the ACE2-low data shown here and the ACE2-high data shown in Fig. S9B. Data shown are representative of two or three independent experiments performed by two different people with similar results.

variants, including the VOCs Alpha and Beta, have significantly higher propensity to adopt "RBD-up" or open state than an early isolate does (49, 61). These data indicate that

Alpha variant is able to utilize cattle and pig ACE2 proteins much more efficiently than the WHU01 strain does. Whether the Alpha variant RBD's markedly increased affinity to cattle and pig ACE2 proteins has biological significance *in vivo* is still to be validated. Nevertheless, considering that cattle and pigs are extremely important livestock animals which serve as major sources of meat for humans, it is necessary in the future to perform *in vivo* studies to evaluate the susceptibility of these species to SARS-CoV-2 variants, especially the widely spreading VOCs.

SARS-CoV-2 Omicron lineage, which was first detected in South Africa and Botswana in early November 2021, has been spreading across the world faster than any previous variants (24). Here, we found that RBDs of WHU01, Alpha, Beta, Gamma, and Omicron showed substantial loss of affinity to increasing number of animal-ACE2 orthologs (Fig. 2, 4, and 6). Omicron RBD showed markedly decreased or complete loss of affinity to multiple tested animal ACE2 orthologs, including that of some livestock animals, pet animals, and wild animals (Fig. 7; Fig. S6 to S8). These data suggest that SARS-CoV-2 might have been getting less infectious to more and more animals as it was evolving in the human populations. However, we also found that RBDs of Alpha, Beta, Gamma, and Omicron have increasingly high affinity to mouse ACE2 and that Omicron RBD mutations are optimal for electrostatic interactions with mouse ACE2 (Fig. 5 and 6). Consistent with this, these variants have been demonstrated able to infect mice (58, 79, 80). Mice are vaccine-inaccessible rodent species that have large population size and could access the territories of both humans and domestic animals. These findings suggest that wild rodents have the potential to become another SARS-CoV-2 reservoir, which could provide a totally independent evolutionary route with selection pressures greatly different from that in human populations.

Perhaps what is more important is that changes in animal ACE2 usage and immune evasion are likely to be highly linked events for SARS-CoV-2. This is because the RBM of the Spike comprises two partially overlapping immunodominant antigenic sites targeted by neutralizing antibodies (81, 82). Consistent with this, three of four RBD mutations strongly favorable for binding mouse ACE2 (Q493R, Q496S, and Q498R; Fig. 5E through H; Fig. S5A) and another three RBD mutations weakly favorable for binding mouse ACE2 (K417N, T478K, and E484A; Fig. 5E; Fig. S5A) also confer immune evasion from anti-SARS-CoV-2 neutralizing antibodies (23, 73, 74). Therefore, if wild animals (e.g., mice) become SARS-CoV-2/Omicron reservoirs, future SARS-CoV-2 variants emerging from these populations will have high probability to be "dual-potential," meaning simultaneously altering animal-ACE2 usage and causing immune escape in humans and thus likely be more dangerous. With hundreds of millions of confirmed SARS-CoV-2 infections in the human populations and potentially also a huge number of SARS-CoV-2 natural infections in farmed or wild animals (30, 41, 42, 83, 84), cross-species receptor usage-directed positive selections will likely play more and more important roles in SARS-CoV-2 evolutions. It is therefore critical to closely monitor animal receptor usage or host range of any widely circulating SARS-CoV-2 variants emerging in the future.

## Limitations of study

SARS-CoV-2 Spike-pseudotyped reporter viruses, instead of live viruses, were used here to study animal ACE2-mediated SARS-CoV-2 entry. Utilization of pseudotyped reporter virus to study virus-receptor interaction and receptor-mediated viral entry is a well-established and widely used method in the field for studying various enveloped viruses, such as HIV (85, 86), influenza virus (87–89), and coronaviruses including SARS-CoV-2 (23, 30, 47, 48, 90–94). This is because findings about virus-receptor interaction based on pseudovirus systems normally correlate well with that based on live viruses, as was exemplified in a number of SARS-CoV-2 studies (30, 93, 94). In addition, all the key findings that are associated with pseudovirus infection assays have been pre-validated in this study using cell surface binding and flow cytometry analysis,

as well as surface plasmon resonance-based quantitative binding kinetics assays. These multi-level validations further offset the limitation of using pseudoviruses.

## MATERIALS AND METHODS

### Cells

293T cells and HeLa cells were kindly provided by Stem Cell Bank, Chinese Academy of Sciences, confirmed mycoplasma free by the provider, and maintained in Dulbecco's Modified Eagle Medium (DMEM, Life Technologies) at 37°C in a 5% $CO_2$-humidified incubator. Growth medium was supplemented with 2 mM Glutamax-I (Gibco, catalog no. 35050061), 100 µM non-essential amino acids (Gibco, catalog no. 11140050), 100 U/mL penicillin and 100 µg/mL streptomycin (Gibco, catalog no. 15140122), and 10% heat-inactivated FBS (Gibco, catalog no. 10099141C). HeLa-based stable cells expressing human ACE2 were maintained under the same culture condition as HeLa, except that 3 µg/mL of puromycin was added to the growth medium. 293F cells for recombinant protein production were generously provided by Dr. Yu J. Cao (School of Chemical Biology and Biotechnology, Peking University Shenzhen Graduate School) and maintained in SMM 293-TII serum-free medium (Sino Biological, catalog no. M293TII) at 37°C, 8% $CO_2$, in a shaker incubator at 125 rpm.

### Plasmids

DNA fragment encoding SARS-CoV-2 Spike variants were synthesized by the Beijing Genomic Institute or Shanghai Sangong Biotech and then cloned into pcDNA3.1(+) plasmid between EcoRI and XhoI restriction sites. Plasmids encoding SARS-CoV-2 Spike variants were generated using an In-Fusion Cloning kit (Takara Bio). The Spike sequences for WHU01 and all the variants investigated in this study were prepared in two different formats, with one containing the natural furin cleavage site between the S1 and S2 fragments and the other having the furin cleavage site motif PRRA deleted (ΔPRRA) as previously described (30). We had previously demonstrated that the ΔPRRA mutation does not affect SARS-CoV-2's sensitivity to neutralizing antibodies or binding affinities to animal ACE2 orthologs (30). The retroviral reporter plasmid encoding a Gaussia luciferase reporter gene was constructed by cloning the reporter genes into the pQCXIP plasmid (Clontech). DNA fragments encoding N-terminally S-tagged ACE2 orthologs were cloned into pQCXIP plasmid (Clontech) between SbfI and NotI restriction sites. Plasmids encoding soluble ACE2 orthologs (18–740 aa) and dimeric RBD variants were generated by cloning each of the gene fragments into a pCAGGS-based human IgG1 or mouse-IgG2a Fc fusion protein expression plasmid between NheI and XhoI sites. Plasmids for monomeric RBD with additional C-terminal HRV 3C protease cleavage site were cloned into the pCAGGS-based human IgG1 Fc fusion protein expression plasmid between NheI and XhoI sites.

### Production and purification of ACE2-huFc, dimeric RBD (RBD-huFc/mFc), or monomeric RBD proteins

293F cells at the density of $6 \times 10^5$ cells/mL were seeded into 100-mL SMM 293-TII serum-free medium (Sino Biological, catalog no. M293TII) 1 day before transfection. Cells were then transfected with 100 µg of ACE2 or RBD expression plasmid in complex with 250-µg polyethylenimine (PEI) MAX 4000 (Polysciences, Inc., catalog no. 24765-1). Cell culture supernatants were collected at 48 to 72 hours post transfection. IgG Fc-containing fusion proteins were purified using Protein A Sepharose CL-4B (GE Healthcare, catalog no. 17-0780-01), eluted with 0.1 M citric acid at pH 4.5, and neutralized with 1 M Tris-HCl at pH 9.0. Buffers were then exchanged to DPBS, and proteins were concentrated by 30-kDa cut-off Amicon Ultra-15 Centrifugal Filter Units (Millipore, catalog no. UFC903096). RBD-Fc proteins containing an HRV 3C protease cleavage site between RBD and Fc were purified using Protein A Sepharose CL-4B and then in-column cleaved by

His-tagged HRV 3C Protease (Thermo, catalog no. 88946) at 4°C for at least 16 hours to release the RBD monomers. His-tagged HRV 3C protease was then removed by HisPur Ni-NTA Resin (Thermo, catalog no. 88221). Buffers were then exchanged to DPBS, and proteins were concentrated by 10-kDa cut-off Amicon Ultra-15 Centrifugal Filter Units (Millipore, catalog no. UFC901096).

## Flow cytometry for detecting interactions of RBD-Ig proteins with cell surface ACE2 orthologs

293T cells in each well of 12-well plates were transfected with 0.4 µL of Lipofectamine 2000 (Life Technologies, catalog no. 11668019) in complex with 125 ng of a plasmid encoding an ACE2 ortholog or its mutant and detached with 5 mM EDTA (Life Technologies, catalog no. 15575020) at 36 hours post transfection. Cells were then stained with 2 µg/mL rabbit anti-S-tag IgG polyclonal antibody (Abcam, catalog no. ab183674) at 37°C for 30 min, followed by washing with PBS and staining with 5 µg/mL RBD-huFc proteins at 37°C for 15 min. Then, cells were stained with 2 µg/mL Alexa488-conjugated goat anti-human IgG secondary antibody (Invitrogen, catalog no. A11013) and Alexa568-conjugated goat anti-rabbit IgG secondary antibody (Invitrogen, catalog no. A11011) at room temperature for 20 min. After washing with PBS, cells were fixed with 0.1% paraformaldehyde–PBS and analyzed using Attune NxT flow cytometer (Thermo Fisher), and data were collected with Attune NxT Software v 4.2 software. Signals of 2,400 ACE2 low-gated cells were collected for each sample.

## SPR assay

The SPR assays were performed at 25°C on a Biacore 8K High-throughput Intermolecular Interaction Analysis System. ACE2-huFc constructs were diluted to 10 µg/mL in 1× assay buffer containing 150 mM NaCl, 0.05% Tween-20, and 10 mM HEPES (pH 7.4). ACE2-huFc was captured to a Protein A sensorchip (GE Healthcare) to a level of 300–700 response units. RBD monomer constructs were serially diluted to 2,000 nM, 1,000 nM, 500 nM, 250 nM, 125 nM, and 62.5 nM in the 1× assay buffer. Each experiment group was followed by cycles of capture for 45 seconds, contact for 120 seconds, dissociation for 300 seconds, and regeneration for 120 seconds. The resulting data were fitted to a 1:1 binding model using Biacore Evaluation Software.

## BLI assay

The BLI assays were performed on a Fortebio Octet RED384 instrument, with the temperature and shaking speed at 30°C and 1,000 rpm, respectively. ACE2-huFc constructs were diluted to 5 µg/mL in 1× assay buffer containing 150 mM NaCl, 0.1% Tween-20, 10 mM HEPES, and 0.1% bovine serum albumin (BSA) (pH 7.4) and used as ligands for the assays. RBD-mFc constructs were serially diluted to 100 nM, 50 nM, 25 nM, 12.5 nM, and 6.25 nM in the 1× assay buffer. Each experiment group started with a 10-min warm-up for pre-hydration of AHC biosensors, followed by cycles of baseline for 60 seconds, loading for 60 seconds, baseline again for 60 seconds, association for 100 seconds, dissociation for 600 seconds, and regeneration plus neutralization for 30 seconds. A 1:1 Langmuir binding model was applied for data processing. All fitted diagrams (global fit) display the entire association window and the first 200 seconds (or 100 seconds only for house mouse and Chinese rufous horseshoe bat ACE2-related assays) of dissociation phase. The BLI sensorgrams were recorded by Octet Data Acquisition 12.0 software and were fitted using Octet Data Analysis HT 12.0 software.

## Western blot to detect S-tagged ACE2 expression in 293T cells

293T cells at 30% density in each well of 96-well plates were reverse transfected with 60 ng of plasmid in complex with 0.15 µL of Lipofectamine 2000 (Life Technologies, catalog no. 11668019). Twenty-four hours after transfection, cells in each well were

lysed with 40-µL lysis buffer, and 5 µL of the lysate was used for Western blot. ACE2 expression was detected using a mouse anti-S-tag monoclonal antibody 6.2 (Invitrogen, catalog no. MA1-981) and a horseradish peroxidase (HRP)-conjugated goat anti-mouse IgG Fc secondary antibody (Invitrogen, catalog no. 31437). Mouse anti-beta-actin IgG monoclonal antibody BA3R (Invitrogen, MA5-15739) was used as a loading control.

## Production of reporter retroviruses pseudotyped with SARS-CoV-2 Spike variants

MLV retroviral vector-based SARS-CoV-2 Spike pseudotypes were produced according to our previous study (30) with minor changes. In brief, 293T cells were seeded at 30% density in 150-mm dishes at 12–15 hours before transfection. Cells were then transfected with 67.5 µg of PEI Max 40,000 (Polysciences, Inc., catalog no. 24765-1) in complex with 3.15 µg of plasmid encoding a Spike variant, 15.75 µg of plasmid encoding murine leukemia virus (MLV) Gag and Pol proteins, and 15.75 µg of a pQCXIP-based luciferase reporter plasmid. Eight hours after transfection, cell culture medium was refreshed and changed to growth medium containing 2% FBS (Gibco, catalog no. 10099141C) and 25 mM HEPES (Gibco, catalog no. 15630080). Cell culture supernatants were collected at 36–48 hours post transfection, spun down at 3,000 × $g$ for 10 min, and filtered through 0.45-µm filter units to remove cell debris. SARS-CoV-2 Spike-pseudotyped viruses were then concentrated 10 times at 2,000 × $g$ using 100-kDa cut-off Amicon Ultra-15 Centrifugal Filter Units (Millipore, catalog no. UFC910024) and subjected to titration and infection assays.

## Pseudovirus titration

Pseudovirus titer were determined using a reverse transcriptase activity assay. Reverse transcriptase-containing pseudoviral particles and recombinant reverse transcriptase standard of known concentrations (TAKARA, catalog no. RR047A) were 10 times diluted with nuclease-free water (Invitrogen, catalog no. 10977015) and lysed with 2× concentrated lysis buffer (0.25% Triton X-100, 50 mM KCL, 100 mM Tris-HCl pH 7.4, 40% glycerol, 1/50 vol of Rnase inhibitor; NEB, catalog no. M0314S) at room temperature for 10 min. Reverse transcription was performed according to the manufacturer's protocol (TAKARA, catalog no. RR047A) using 1 µL of the lysate as reverse transcriptase and TRIzol reagent-isolated 293T total RNA as template. Reverse transcription products were then subjected to qPCR with a commercial kit (TAKARA, catalog no. RR820Q) to amplify GAPDH (forward primer: 5′-CCACTCCTCCACCTTTGAC-3′, reverse primer: 5′-ACCCTGTTGCTGTAGCCA-3′) in Applied Biosystems QuantStudio 5. Standard curves were generated based on qPCR cycle threshold (Ct) values obtained with serially diluted recombinant reverse transcriptase standard.

## SARS-CoV-2 pseudovirus infection of 293T cells expressing ACE2 orthologs

Pseudovirus infection assay was performed according to our previous study (30) with minor modification. In brief, 293T cells at 30% density in each well of gelatin (Millipore, catalog no. ES-006-B) pre-coated 96-well plates were reverse transfected with 0.15 µL of Lipofectamine 2000 (Life Technologies, catalog no. 11668019) in complex with 60 ng of a vector control plasmid or a plasmid encoding an ACE2 ortholog or its mutant. Twenty-four hours post transfection, cells in each well were infected with SARS-CoV-2 pseudoviral particles equivalent to 8 × $10^{10}$ U reverse transcriptase. Cell culture supernatants were collected and subjected to a Gaussia luciferase assay at 48 hours post infection.

## SARS-CoV-2 pseudovirus neutralization assay

Pseudovirus neutralization experiments were performed following our previous study (30). In brief, SARS-CoV-2 Spike variant-pseudotyped luciferase reporter viruses were pre-diluted in DMEM (2% FBS, heat inactivated) containing titrated amounts of an

ACE2-huFc construct. Virus inhibitor mixtures were incubated at 37°C for 30 min and then added to HeLa-hACE2 cells in 96-well plates and incubated overnight at 37°C. Virus inhibitor-containing supernatant was then removed and changed with 150 µL of fresh DMEM (2% FBS) and incubated at 37°C. Culture medium was refreshed every 12 hours. Cell culture supernatants were collected for Gaussia luciferase assay at 48 hours post infection.

## Gaussia luciferase luminescence flash assay

To measure Gaussia luciferase expression, 20 µL of cell culture supernatant of each sample and 100 µL of assay buffer containing 4 µM coelenterazine native (Biosynth Carbosynth, catalog no. C-7001) were added to one well of a 96-well black opaque assay plate (Corning, catalog no. 3915) and measured with Centro LB 960 microplate luminometer (Berthold Technologies) for 0.1 second/well.

## Molecular dynamics simulation and MM/GBSA analysis

To build the initial model of RBD:ACE2 complexes, we selected representative crystal structures as the templates, i.e., WHU01-RBD: human ACE2 complex (PDB ID: 6LZG), Omicron-RBD: human ACE2 complex (PDB ID: 7WBP), K417-N501Y-RBD: mouse ACE2 complex (PDB ID: 7FDK), and Omicron RBD: mouse ACE2 complex (PDB ID: 7XO6). N-glycosylation on N343 (95) of RBD is set up via CHARMM-GUI (96), as is detailed in our latest MD study (72). The protein and glycans are using CHARMM36m (97) force field, where we further test CHARMM36 (98) force field on Omicron-RBD: human ACE2 complex to ensure that the choice of force field does not affect the relative stability of key salt bridges. The protein complexes are solvated in a cubic solvent box with TIP3P water (99) molecule layer extended approximately 10 Å away from the surface of the proteins. Counter ions (K+ and Cl−) were randomly placed to ensure electrostatic neutrality corresponding to an ionic concentration of ~130 mM. The LINCS (100) algorithm was used to restrict the covalent bond including hydrogen atom and increase the integration time step to 2 fs, and the particle-mesh Ewald method (101) was used to calculate the long-range electrostatic interactions with a real-space cutoff of 12 Å.

In simulation stage, the backbone atoms were restrained to the crystal coordinates, and the system was minimized with 5,000-step steepest descent minimization method to reduce local high-energy contacts. The system was gradually heated to 300 K during 1-ns NVT simulation with harmonic restraints of 1,000 kJ/(mol*Å$^2$) on heavy atoms of both proteins. In the following 5-ns simulation, the harmonic restrains were gradually released under NPT ensemble with 1 atm. Then, each system was simulated at least 10 ns to further equilibrate. Finally, the MD simulations were extended for 100 to 200 ns and even longer. All simulations are performed simultaneously using GROMACS 2019 (102). Based on the MD simulation trajectory, we use the MM/GBSA (103) technique to quickly calculate residue-specific binding energy at the interface. According to the thermodynamic cycle, the binding energy is divided into molecular mechanics term in vacuum and solvation energy term; the latter can be further divided into polar solvation energy calculated by generalized born function and nonpolar solvation energy that is proportional solvent-accessible surface area in MM/GBSA. This decomposition scheme provides the contribution of each residue to the binding energy. In this study, we performed MM/GBSA analysis on the last 100-ns MD trajectories (2,000 snapshots) via gmx-MMPBSA (104). The contribution of all amino acids within 10 Å from the binding interface to the binding energy was collected.

## Experimental data collection and analysis

All the experiments were independently performed for two or three times by two different people. Flow cytometry data were analyzed by FlowJo V10 software. The SPR data were fitted to a 1:1 binding model using Biacore Evaluation Software. Image

Lab Software (Bio-Rad) was used to collect SDS-PAGE and Western blot image data. MikroWin 2000 Software (Berthold Technologies) was used to collect luciferase assay data. The BLI sensorgrams were recorded by Octet Data Acquisition 12.0 software and were fitted using Octet Data Analysis HT 12.0 software. ACE2 sequence alignment were performed with BioEdit Sequence Alignment Editor. Structure models of animal ACE2 proteins were generated using SWISS-MODEL protein structure homology modeling server (67). Structure superimposition was performed using the MatchMaker tool of the UCSF Chimera software (68). GraphPad Prism 6.0 software was used for preparation of bar graphs and neutralization curves and statistical analyses. Adobe Illustrator 2022 was used for preparation of the main figures (Fig. 1 to 7) and supporting figures (Fig. S1 to S9) for the manuscript.

## Statistical analysis

Data shown in the figures are representative of two or three independent experiments performed by two different people with similar results, and data points represent mean values ± SD of three biological replicates.

## ACKNOWLEDGMENTS

We thank the Biochemistry Core of the Shenzhen Bay Laboratory (Shenzhen, China) for providing help on performing the flow cytometry, SPR- and BLI-based binding kinetics assays, and data analysis. We thank Dr. Yu J. Cao (School of Chemical Biology and Biotechnology, Peking University Shenzhen Graduate School, Shenzhen, China) for providing the 293F cells used in this study for the production of recombinant proteins.

This work was supported by Shenzhen Bay Laboratory Major Program of Shenzhen Bay Laboratory (S201101001-2, G.Z. and Y.L.), Key COVID-19 Program of Shenzhen Bay Laboratory (S211410002, G.Z. and Y.L.), Shenzhen Municipal Science and Technology Innovation Commission (KQTD2017-0330155106581 to J.G.), Guangdong Pearl River Talent Program (2021QN02Y618 to Y.W.), the National Natural Science Foundation of China (92269102, 22007069 to Y.W.), and the Natural Science Foundation of HLJ (LH2019C086 to D.C.).

## AUTHOR AFFILIATIONS

[1]School of Chemical Biology and Biotechnology, Peking University Shenzhen Graduate School, Shenzhen, China
[2]Shenzhen Bay Laboratory, Shenzhen, China
[3]Hubei JiangXia Laboratory, Wuhan, Hubei, China
[4]NHC Key Laboratory of Hormones and Development, Tianjin Key Laboratory of Metabolic Diseases, Chu Hsien-I Memorial Hospital & Tianjin Institute of Endocrinology, Tianjin Medical University, Tianjin, China
[5]Horae Gene Therapy Center, University of Massachusetts Chan Medical School, Worcester, Massachusetts, USA
[6]Heilongjiang Academy of Medical Sciences, Harbin, China
[7]Biomedical Research Center of South China, Fujian Normal University, Fuzhou, China
[8]Center for Infectious Disease Research, School of Medicine, Tsinghua University, Beijing, China
[9]Department of Chemistry and Supercomputing Institute, University of Minnesota, Minneapolis, Minnesota, USA

## PRESENT ADDRESS

Guocai Zhong, Horae Gene Therapy Center, University of Massachusetts Chan Medical School, Worcester, Massachusetts, USA
Guocai Zhong, RNA Therapeutics Institute, University of Massachusetts Chan Medical School, Worcester, Massachusetts, USA

Guocai Zhong, Department of Biochemistry and Molecular Biotechnology, University of Massachusetts Chan Medical School, Worcester, Massachusetts, USA

## AUTHOR ORCIDs

Weitong Yao  http://orcid.org/0000-0001-5441-9120
Yujun Li  http://orcid.org/0000-0002-1011-0452
Danting Ma  http://orcid.org/0009-0001-3386-2585
Qiang Ding  http://orcid.org/0000-0002-6226-869X
Yingjie Wang  http://orcid.org/0000-0001-9800-8163
Jiali Gao  http://orcid.org/0000-0003-0106-7154
Guocai Zhong  http://orcid.org/0000-0002-7609-9575

## FUNDING

| Funder | Grant(s) | Author(s) |
|---|---|---|
| Shenzhen Bay Laboratory (SZBL) | S201101001-2, S211410002 | Guocai Zhong |
| Shenzhen Bay Laboratory (SZBL) | S201101001-2, S211410002 | Yujun Li |
| Shenzhen Municipal Science and Technology Innovation Council | KQTD2017-0330155106581 | Jiali Gao |
| Guangdong Pearl River Talent Program | 2021QN02Y618 | Yingjie Wang |
| National Natural Science Foundation of China (NSFC) | 92269102 | Yingjie Wang |
| 黑龙江省科技厅 \| Natural Science Foundation of Heilongjiang Province (Heilongjiang Natural Science Foundation) | LH2019C086 | Dechun Cheng |

## AUTHOR CONTRIBUTIONS

Weitong Yao, Data curation, Formal analysis, Investigation, Visualization, Writing – original draft, Writing – review and editing | Yujun Li, Data curation, Formal analysis, Funding acquisition, Investigation, Visualization, Writing – original draft, Writing – review and editing | Danting Ma, Data curation, Formal analysis, Investigation, Visualization, Writing – review and editing | Xudong Hou, Data curation, Investigation, Visualization | Haimin Wang, Investigation, Writing – original draft, Writing – review and editing | Xiaojuan Tang, Investigation | Dechun Cheng, Funding acquisition, Investigation, Methodology | He Zhang, Investigation | Chengzhi Du, Investigation | Hong Pan, Investigation | Chao Li, Methodology | Hua Lin, Investigation, Methodology | Mengsi Sun, Methodology | Qiang Ding, Methodology | Yingjie Wang, Data curation, Formal analysis, Funding acquisition, Investigation, Visualization, Writing – original draft, Writing – review and editing | Jiali Gao, Formal analysis, Funding acquisition, Writing – original draft, Writing – review and editing | Guocai Zhong, Conceptualization, Formal analysis, Funding acquisition, Visualization, Writing – original draft, Writing – review and editing

## DATA AVAILABILITY

The study did not generate unique datasets or code. All relevant data and methods information have been provided in the manuscript and its supplemental file. All reagents developed in this study, such as vector plasmids, will be made available upon written request.

## ADDITIONAL FILES

The following material is available online.

## Supplemental Material

**Supplemental file 1 (Spectrum02676-23-s0001.pdf).** Fig. S1 to S9 and Tables S1 to S5.

## Open Peer Review

**PEER REVIEW HISTORY (review-history.pdf).** An accounting of the reviewer comments and feedback.

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
