## [Reviewer comments · Microbiology Spectrum]

Microbiology Spectrum

Evolution of SARS-CoV-2 Spikes shapes their binding affinities to animal ACE2 orthologs

Weitong Yao, Yujun Li, Danting Ma, Xudong Hou, Haimin Wang, Xiaojuan Tang, Dechun Cheng, He Zhang, Chengzhi Du, Hong Pan, Chao Li, Hua Lin, Mengsi Sun, Qiang Ding, Yingjie Wang, Jiali Gao, and Guocai Zhong

Corresponding Author(s): Guocai Zhong, University of Massachusetts Chan Medical School

Review Timeline:

Submission Date:	July 10, 2023
Editorial Decision:	September 6, 2023
Revision Received:	October 4, 2023
Accepted:	October 8, 2023

Editor: Tao Deng

Reviewer(s): The reviewers have opted to remain anonymous.

Transaction Report:

DOI: <https://doi.org/10.1128/spectrum.02676-23>

September 6, 2023

Dr. Guocai Zhong
University of Massachusetts Chan Medical School
Biochemistry and Molecular Biotechnology
364 Plantation Street LRB-815
Worcester, MA 01605

Re: Spectrum02676-23 (Evolution of SARS-CoV-2 Spikes shapes their binding affinities to animal ACE2 orthologs)

Dear Dr. Guocai Zhong:

Thank you for submitting your manuscript to Microbiology Spectrum. As you will see your paper is very close to acceptance. Please modify the manuscript along the lines I have recommended. As these revisions are quite minor, I expect that you should be able to turn in the revised paper in less than 30 days, if not sooner. If your manuscript was reviewed, you will find the reviewers' comments below.

When submitting the revised version of your paper, please provide (1) point-by-point responses to the issues raised by the reviewers as file type "Response to Reviewers," not in your cover letter, and (2) a PDF file that indicates the changes from the original submission (by highlighting or underlining the changes) as file type "Marked Up Manuscript - For Review Only". Please use this link to submit your revised manuscript. Detailed instructions on submitting your revised paper are below.

Link Not Available

Sincerely,

Tao Deng

Reviewer comments:

Reviewer #1 (Comments for the Author):

Weitong et al investigated the binding affinities of spike protein to ACE2 orthologs of 18 animal species. This is a well-written paper with solid data support. The results in the paper help us understand the impact of spike mutations on ACE2 binding across mammalian species.

Major comments:

In Introduction, authors should briefly review surveillance studies of SARS-COV-2 in animal species, such as strains found in cats and dogs, zoo animals, and minks in farm. What animals can harbor SARS-COV-2 as carrier? Review previous studies of RBD-ACE2 binding in animals, and how this paper is distinct from other studies.

How many replicates of measurements were performed for each data point in Figure 2C/D, is it possible some differences are due to experimental variation or batch effect? It seems all RBDs failed to bind rat ACE2 (no difference) compared to no RBD in ACE2 low cells, how to explain this (similar for civet)?

Authors may explain better in Fig. 2 C/D, Fig 7B/D what implications of ACE2-high versus ACE2-low population data, are they equally important? How to explain the discordances between the two populations such as rat?

Can authors comment or predict which animal's ACE2 binding profile could be similar to that of minks that suffered from large SARS-COV-2 outbreaks in European farms (based on model prediction or literature).

Minor comments:

1. Line 64-65, progresses have been achieved in developing prophylactic vaccines and antibody therapeutics (1-13). There are small molecule drugs like Paxlovid against SARS-COV-2, please revise and add references.
2. Line 74, according to Organization (WHO) online updates, need to include web link as citation.
3. Line 82, SARS-CoV-2 utilizes ACE2 as an essential cellular receptor (26, 32, 33). Suggest adding sentences about RBD here, where it is located, its significance and how RBD interacts with ACE2.
4. How to define ACE2 high and low cells? <1000 MFI is low and >4000 as high?
5. Line 119-20, to more and more ACE2. What does this mean? revise.

Preparing Revision Guidelines

Please return the manuscript within 60 days; if you cannot complete the modification within this time period, please contact me. If you do not wish to modify the manuscript and prefer to submit it to another journal, please notify me of your decision immediately so that the manuscript may be formally withdrawn from consideration by Microbiology Spectrum.

Response to Reviewers (reviewer comments in black; author responses in blue)

Reviewer #1 (Comments for the Author):

Weitong et al investigated the binding affinities of spike protein to ACE2 orthologs of 18 animal species. This is a well-written paper with solid data support. The results in the paper help us understand the impact of spike mutations on ACE2 binding across mammalian species.

We appreciate the reviewer's careful evaluation and positive comments on the quality and significance of our study.

Major comments:

1. In Introduction, authors should briefly review surveillance studies of SARS-COV-2 in animal species, such as strains found in cats and dogs, zoo animals, and minks in farm. What animals can harbor SARS-COV-2 as carrier? Review previous studies of RBD-ACE2 binding in animals, and how this paper is distinct from other studies.

We thank the reviewer for these comments. As suggested, we have added to the Introduction section a brief review of the studies of SARS-COV-2 in animal species (lines 85-94).

Regarding the comment - 'review previous studies of RBD-ACE2 binding in animals, and how this paper is distinct from other studies', such a review had previously been included in the first paragraph of the Discussion section. We've kept the review and introduced a minor revision as shown in the revised main text lines 312-317: *'Similar studies involving a panel of animal ACE2 orthologs have previously only been performed for the prototype SARS-CoV-2 Spike (30, 31, 33, 47). Since receptor usage is a critical determinant for the host range of coronaviruses (32), the current study concerning 108 distinct interactions (18 animal ACE2 orthologs × 6 major SARS-CoV-2 Spike variants) provides more comprehensive and useful information for understanding possible cross-species transmission risks of all the five VOCs'*. Note that one of the previous studies of RBD-ACE2 binding in animals is our own study, which describes the interactions of 4 related CoVs (SARS-CoV-2, SARS-CoV-1, pangolin-CoV, bat-CoV RaTG13) with 15 animal ACE2 orthologs (Li et al. *SARS-CoV-2 and Three Related Coronaviruses Utilize Multiple ACE2 Orthologs and Are Potently Blocked by an Improved ACE2-Ig*. *J Virol*. 2020. PMID: 32847856).

2. How many replicates of measurements were performed for each data point in Figure 2C/D, is it possible some differences are due to experimental variation or batch effect?

We thank the reviewer for careful evaluation of our data. The experiments in Figure 2C/D were repeated twice with highly similar results. However, some very minor MFI differences might indeed come from experimental variation or batch difference, for example the MFI differences between the Delta/donkey (MFI: 2013 for ACE2-high, 213 for ACE2-low) vs Delta/goat (MFI: 2008 for ACE2-high, 225 for ACE2-low) interactions. Thus, in the Figures 3-7, we mainly focus on changes that are (1) more significant and (2) reproducible in both ACE2-high and ACE2-low cells in Figure 2. In addition, all the key findings from this flow cytometry analysis have then been further validated using surface plasmon resonance (SPR)-

or Bio-layer interferometry (BLI)-based, more accurate, and quantitative binding kinetics studies (Fig 3B, 3E-H, 4B, 5B, 5G, 7A, 7C, S3D and S4B), mutagenesis studies (Fig 3E-H, 4C-H, 5E-H, 7B-D), and pseudovirus infection studies (Fig 3C, 4E, 4G, 5C, 5D, 5H, S6B). We believe that these multi-level validations offset the limitation/minor variations of the data in Figure 2C/D. This is also discussed in the Limitation of study section followed by the Discussion section.

3. It seems all RBDs failed to bind rat ACE2 (no difference) compared to no RBD in ACE2 low cells, how to explain this (similar for civet)?

First, the finding in Figure 2C/D that WHU01 RBD fails to bind rat Ace2 is consistent with the rat-Ace2 binding data in our previous study (*Li et al. SARS-CoV-2 and Three Related Coronaviruses Utilize Multiple ACE2 Orthologs and Are Potently Blocked by an Improved ACE2-Ig. J Virol. 2020. PMID: 32847856*). Second, we can see significant binding of rat Ace2 to Alpha, Beta, and Gama RBDs in the ACE2-high cells (Fig 2C). Third, although the MFI signal of Alpha/Beta/Gama RBD binding to rat Ace2 looks weak in Figure 2D, the flow cytometry dot plot raw data in Figure S2 clearly show that Alpha, Beta, and Gama RBDs do bind rat Ace2. The following are high resolution images from Figure S2. The magenta boxes mark the RBD-positive cell populations in each sample. You can see rat Ace2 and RBD double positive cell populations in Alpha, Beta, and Gama RBD staining samples. Finally, these double-positive cell populations are so small in the rat Ace2-low cells that the binding can't be significantly reflected by MFI. This is likely because the interactions observed in ACE2-high cells are actually low-affinity interactions that rely on the avidity effect of high ACE2 density on the cell surface. Because of this reason, we think the interactions observed in the ACE2-low cells are more likely to be high-affinity interactions and thus more likely to be physiologically relevant. We therefore opt to focus more on the interactions or changes observed in the ACE2-low cells in the subsequent part of this study (Figs 3-7).

Likely for the same reasons, interactions of civet Ace2 with multiple tested Spike variants are less significant/visible in ACE2-low cells (Fig 2D) than in ACE2-high cells (Fig 2C).

4. Authors may explain better in Fig. 2 C/D, Fig 7B/D what implications of ACE2-high versus ACE2-low population data, are they equally important? How to explain the discordances between the two populations such as rat?

We thank the reviewer for this comment. As described above in the response to the reviewer's previous comment, the interactions observed in the ACE2-low cells are more likely to be high-affinity interactions and thus more likely to be physiologically relevant. We therefore should focus more on the interactions or changes observed in the ACE2-low cells in the subsequent part of this study (Figs 3-7). We have now added a sentence to introduce this point in the main text (lines 152-155).

Regarding 'the discordances between the two populations such as rat', please refer to the

above detailed response to the reviewer's major comment #3.

5. Can authors comment or predict which animal's ACE2 binding profile could be similar to that of minks that suffered from large SARS-COV-2 outbreaks in European farms (based on model prediction or literature).

Indeed, it will be very interesting to know if any other animal may also have high risk of human-animal-animal-human transmission similar to that of minks. Apparently, this is an interesting open question. Although it is beyond the scope of our study, based on our dataset, we think continuous surveillance of mouse samples for possible SARS-CoV-2 infection/evolution would be necessary. For the following reasons, we think mice have the potential to become another SARS-CoV-2 reservoir, which could provide a totally independent evolutionary route with selection pressures greatly different from that in human populations. First, mouse Ace2 showed significant gain of binding to Omicron Spike, instead of loss of binding observed with other orthologs, in the current study. Second, mice are able to support Beta, Gamma, and Omicron infection/replication. Third, mice are vaccine-inaccessible species. Fourth, mice have large population size. Last, mice could access the territories of both humans and domestic animals.

Then, based on literature, white-tailed deer is apparently another animal population that should and have been under continuous surveillance for human to animal, animal to animal, and animal to human SARS-CoV-2 transmissions.

Minor comments:

1. Line 64-65, progresses have been achieved in developing prophylactic vaccines and antibody therapeutics (1-13). There are small molecule drugs like Paxlovid against SARS-COV-2, please revise and add references.

We have revised the text as suggested (lines 64-65) and added three new references (refs 14-16) regarding small molecule anti-SARS-CoV-2 drugs.

2. Line 74, according to Organization (WHO) online updates, need to include web link as citation.

The link has been added accordingly (line 75).

3. Line 82, SARS-CoV-2 utilizes ACE2 as an essential cellular receptor (26, 32, 33). Suggest adding sentences about RBD here, where it is located, its significance and how RBD interacts with ACE2.

We appreciate the reviewer for this very constructive comment. We have added sentences and references as suggested (lines 95-104).

4. How to define ACE2 high and low cells? <1000 MFI is low and >4000 as high?

The gating strategy for defining ACE2-high and ACE2-low cells are shown in Fig S2. Specifically, ACE2-low cells are ACE2 positive cells with ACE2 staining MFI in the range of

2.5×10^3 to 2.5×10^4 , while ACE2 high cells are ACE2 positive cells with ACE2 staining MFI in the range of 2.5×10^2 to 2.5×10^3 . We thank the reviewer for pointing out this. To increase the clarity of the paper, we have introduced the definition and figure S2 citation in the main text (lines 133-135) as well as in the Figure 2 legend (lines 1067-1069).

5. Line 119-20, to more and more ACE2. What does this mean? revise.

We have revised the text in the original lines 119-120 (now line 139), as well as the text with same issue in line 340, to 'loss of affinity to increasing number of animal-ACE2 orthologs'.

October 8, 2023

Dr. Guocai Zhong
University of Massachusetts Chan Medical School
Biochemistry and Molecular Biotechnology
364 Plantation Street LRB-815
Worcester, MA 01605

Re: Spectrum02676-23R1 (Evolution of SARS-CoV-2 Spikes shapes their binding affinities to animal ACE2 orthologs)

Dear Dr. Guocai Zhong:

Your manuscript has been accepted, and I am forwarding it to the ASM Journals Department for publication. You will be notified when your proofs are ready to be viewed.

Sincerely,

Tao Deng
Editor, Microbiology Spectrum
